# Urban Industrial Tourism: Cultural Sustainability as a Tool for Confronting Overtourism—Cases of Madrid, Brussels, and Copenhagen

Carmen Hidalgo-Giralt * , Antonio Palacios-García , Diego Barrado-Timón and José Antonio Rodríguez-Esteban

Department of Geography, Autónoma University of Madrid, 28049 Madrid, Spain; antonio.palacios@uam.es (A.P.-G.); diego.barrado@uam.es (D.B.-T.); josea.rodriguez@uam.es (J.A.R.-E.)
* Correspondence: carmen.hidalgog@uam.es; Tel.: +34-914974587

**Abstract:** The chief objective of this research was to analyze how the industrial heritage of three European capitals—Madrid, Brussels, and Copenhagen—has been integrated into the dynamics of their urban tourism, thereby generating new resources and cultural spaces. In regards to the latter point, this study poses the working hypothesis that industrial heritage can function as a tool for cultural sustainability, which allows for deconcentration away from historic city centers subjected to significant overtourism. To verify this hypothesis, a methodology has been designed based on the selection of specific indicators and the creation of maps, taking as reference data from the Tripadvisor travel portal. The results obtained are truly encouraging, and it would be interesting to expand this study by incorporating new case studies to allow us to discern additional patterns of behavior around urban industrial tourism.

**Keywords:** overtourism; sustainability; industrial heritage; industrial tourism; new urban spaces; cultural resources

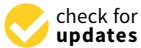



## 1. Introduction

This research springs from a problem widely observed in the international context before the outbreak of COVID-19: an increase in tourist pressure within the historic centers of cities, where the main resources and services for tourism are located, as a consequence of very rapid and unprecedented increases in numbers of visitors. This phenomenon, known as "overtourism" [1,2], responds to multiple and complex causes, including the development of platform economies, the proliferation of new business models based on "peer-to-peer" and blockchain technologies, the diffusion of "low-cost" offers, and the international positioning of cities. These causes converged in the consolidation of a kind of urban tourism promoted by both public and private agents as a means to revitalize cities and their cultural industries [3–5], following the economic recession of 2008, all within a clear context of the promotion of neoliberal policies [6]. The impacts of overtourism are numerous and diverse [7,8], but particularly noteworthy is "touristification", referring to the transformations within historic centers meant to satisfy the needs of tourists, and to the detriment of the requirements of a city's local population [9,10]. The intensification of tourism generates, for example, changes in cultural facilities in order to target tourists, transformations in housing markets that favor proliferation of tourist apartments and the emergence of real estate investment funds, and ultimately, the expulsion of the local population, to be replaced by an extremely volatile tourist flow [11,12]. As a consequence, the historic centers of the cities so affected lose their identity [13], engendering movements of "tourismophobia", which is the perception of tourism as a negative activity for local inhabitants by limiting the "right to the city", as indicated by Levefbre [14–17].

Among the strategies to confront the growing pressure from tourism, sustainable development is understood in this research as the theoretical–practical framework that

should articulate such responses. Defined by the Brutdland Report [18] as a means to "satisfy the needs of present generations without compromising the possibilities of future generations to meet their own needs", and consolidated after the Rio Declaration (1992), sustainable development currently figures in national political agendas through the United Nations 2030 Agenda and its 17 objectives [19]. The concept of sustainability has undergone a significant evolution from its origins in the 1980s to the present day. The contemporary notion of sustainable development is based on three dimensions—the environmental, the ecological, and the social [20]—to which a fourth, the cultural dimension, has sometimes been added in response to the reflections of some authors [21,22].

Attending to the principles of cultural sustainability and the relevance of heritage, this research addresses the theory that, under the umbrella of cultural sustainability, it is possible to create new cultural heritages that contribute to the economic, social, and environmental development of territories [23]. As asserted by Martinell et al. [24], it is important that societies be capable of "identifying and generating cultural heritage, reusing cultural references that a society makes its own, whether because these are useful in transforming areas of life or in overcoming crises of any kind." In short, this is a new strategy that favors the reduction of tourist pressure in urban centers and the decongestion of areas most affected by tourist activity through the creation of new cultural heritage. Furthermore, this strategy allows the development of new tourism models that are more respectful of the social, economic, and environmental dynamics of cities, establishing paradigms of planning and management that are more sustainable [23].

Given the magnitude of the process of enhancing the tourism value of cultural heritage, this research is directed at the representativeness and touristic use of cultural heritage. This process is examined within a specific typology of cultural heritage—industrial heritage—along with the tourism modality of industrial tourism. Numerous cities count old industrial infrastructures among their main tourist resources. Paradigmatic cases include the Tate Modern in London (formerly a power station, now a center for contemporary art of international high standing), the Grande Halle de la Villete in Paris (once an animal market, today a center for various events), and the Buchändlerhof substation in Berlin (which, in the 1990s, became the E-Werk, one of the most famous techno music clubs in Europe).

Based on these examples, the main working hypothesis of this research is that the integration of industrial heritage can diversify the range of tourism products offered by a city and also generate new spaces to help deconcentrate tourist flows away from historical centers. To verify this approach, a methodology has been designed that (through a series of quantitative indicators) permits comparisons of touristic functionality among the industrial heritage sites in three European cities: Copenhagen, Denmark; Brussels, Belgium; and Madrid, Spain. This analysis is complemented by heat-map representations of the areas of influence of these cities' industrial heritage sites, constructed from Tripadvisor ratings by tourists who have visited them, and demonstrating their distance from the historic centers. The results obtained allowed for the determination of which factors are currently contributing to (or hindering) the appreciation of tourists for industrial heritage in these cities, as well as the verification of our hypothesis that industrial heritage can diversify the touristic offerings and, in terms of sustainability, help to cope with touristic intensity.

Industrial heritage is a fairly recent discipline, scarcely 70 years old; while the ability of industrial heritage to attract tourist flows has prompted more interest over the past decade, studies remain few. It is therefore difficult to compare our results with those of similar investigations. Although this research certainly contributes to covering a gap in this area of knowledge, the number of cities analyzed should be increased to obtain more viable patterns of behavior related to urban industrial tourism from a technical/scientific perspective. In addition to this limitation, it must be noted that the COVID-19 pandemic has altered the orientation of this study. This unusual but persistent circumstance has necessarily become a factor in our analysis, given that the scenario created by the pandemic has introduced an element of uncertainty that could conceivably compromise the validity of the data over the long term.

## 2. Literature Review and Theoretical Frame

### 2.1. Strategies to Deal with the Intensification of Tourism, from Sustainable Development to Tourism Decline and Resilience

Tourism—among the fastest-growing economic activities on a global scale in recent decades—is no stranger to concerns around sustainability. The World Tourism Organization (UNWTO) is the UN body in charge of promoting sustainable tourism, defining it as an activity "which takes fully into account both current and future economic, social, and environmental repercussions in meeting the needs of visitors, the industry, the environment, and the host communities". UNWTO articulates sustainable tourism across three fundamental pillars: making optimal use of environmental resources; respecting the cultural authenticity of host communities and ensuring that activities be economically viable over the long term [25]. Cultural tourism—a modality integrating industrial tourism, the object of study here—is also acquainted with this reality. In recent years, cultural tourism has likewise experienced considerable growth, putting the viability of tourist destinations at risk, and moving sectors of society to demand the implementation of sustainable models that are much more respectful of both territories and local communities [25]. Emerging alongside this model of sustainable tourism are other, more critical approaches that cannot be ignored, including the post-growth perspective [26–29], which proposes an "alternative scenario to 'growthism' that advocates a socio-economic, ecological, democratic, and just transition which guarantees the long-term viability of life on the planet" [26]. Additionally, interesting are approaches that relate tourism to territorial resilience [30–34], which is to say "the capacity of a destination to balance and absorb the impacts of crises, taking into account its prior situation (prior resources, organization, structures, and functional adaptability). The more the resistance, the smaller the changes" [35].

### 2.2. Cultural Sustainability: A Tool to Create New Heritages—The Process of Enhancing the Tourist Value of Cultural Heritage

The concept of sustainability has undergone a significant evolution from its origins in the 1980s to the present day. The contemporary notion of sustainable development is based on three dimensions—the environmental, the ecological, and the social [36]—to which a fourth, the cultural dimension, has sometimes been added in response to the reflections of some authors [36,37]. Cultural sustainability is difficult to define, since the terms "culture" and "sustainability" are themselves extremely complex, as are their interrelations [36,37]. Soini and Dessein [37] analyze the meaning of cultural sustainability and indicate three different representations: "Culture *in* Sustainability" (that is, culture as an achievement in development), "Culture *for* Sustainability" (culture as a resource and condition for development), and "Culture *as* Sustainability" (development as a cultural process). By way of a bibliographic analysis of the scientific literature, Soini and Bikerland [38] group publications on cultural sustainability into seven storylines (cultural heritage, cultural vitality, cultural diversity, economic viability, locality, eco-cultural resilience, and eco-cultural civilization), among which the first—cultural heritage—forms the central axis of this research.

Theoretical approaches to cultural heritage from the perspective of cultural sustainability are very diverse. Some authors stress the protection of local and regional cultural identities, as well as their durability over time [39]. Others focus on the effects of globalization, and especially how migratory flows and climate change threaten the cultural identity of peoples [40,41]. Nor can reflections related to the conservation of cultural heritage, urban development, and new technologies be overlooked [17]. While cultural heritage does not exert much weight in the 2030 Agenda, the New Urban Agenda adopted at the Habitat III conference in 2016 recognizes its contribution to the urban development of cities [19], and public and private agents today agree that cultural heritage is the inheritance of a capital that must be passed along to future generations [42].

The progressive attribution of cultural value to elements both physical and intangible, and consequently, their consideration as heritage, is a process of enormous social

and cultural complexity. In turn, the subsequent consideration and enhancement of this heritage as a tourist resource involves complex and expensive procedures that do not always lead to positive outcomes [43]. This process usually begins with a social pressure group demanding the protection and preservation of the exceptional cultural or natural values of a particular resource in the territory, and, depending on the case, this can launch a process of administrative protection of the said asset. In this way, there is a transformation of the territorial element into cultural or natural heritage. Subsequently, this new heritage awakens the interest of the tourist, who begins to visit it, as well as the interest of public administrations, which adapt the heritage to tourism by providing visitors with the necessary infrastructures. At this point, heritage has been transformed into a tourist resource, which, depending on the interest and strength of local agents, public and private, is then integrated into the tourism dynamics and products of a given territory. A certain specialization of destinations can even be generated in this type of heritage (Figure 1).

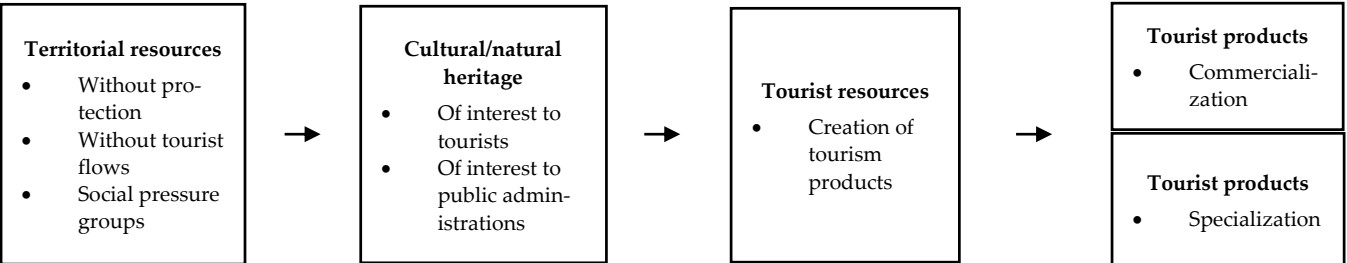

**Figure 1.** Model of the process of enhancing the touristic value of cultural/natural heritage. Source: Own elaboration based on Hidalgo-Giralt [43].

### 2.3. The Transformation of Obsolete Industrial Facilities into Cultural Heritage; New Resources for a New Tourism

The deindustrialization affecting developed countries from the second half of the 20th century led to the abandonment and disaffection of many industrial facilities (complexes, factories, mines, etc.). Unfortunately, many of these old infrastructures—reflections of a bygone era—have vanished, while others have undergone a process of social, cultural, and economic enhancement that has allowed their transformation into cultural heritage. In fact, a new typology of cultural heritage has been created: the industrial, conceived as "remains of industrial culture that have a historical, technological, architectural, or scientific value. These consist of buildings and machinery, workshops, factories and mills, mines and sites for processing and refining, warehouses and deposits, places of energy generation and transmission and use, modes of transport and all their infrastructures, as well as places where social activities related to industry such as housing, religious worship, or education once took place" [44]. Derived from this newly created cultural heritage of industrial origin, a new modality of cultural tourism has also emerged: industrial tourism, which (with nuances) [45] consists of visits to old industrial facilities [44]. In fact, there are numerous industrial regions (such as the Ruhr in Germany, Montceau-les-Mines in France, or Coal-Brookdale in the United Kingdom) that have opted for the touristic enhancement of their industrial heritage as a tool with which to confront the economic, social, and cultural crises derived specifically from the closure of industrial facilities.

On an urban scale, the process has been similar. Post-industrial cities have had to reinvest in order to position themselves nationally or internationally [46], and the cultural heritage of cities has been likewise enriched through the reconstitution of old industrial facilities [47–53]. However, unlike in industrial regions, industrial tourism in cities has not been fully consolidated as a consequence of the dearth of tourism products articulated through the rationale of industrial heritage, their scant integration into the city's tourism planning, and a general lack of knowledge around this type of tourist attraction [54,55].

## 3. Materials and Methods

### 3.1. Hypothesis, Objectives, and Work Methodology

With consideration given to the statements in the previous section, the main working hypothesis of this research is that the transformation of old urban industrial facilities into cultural spaces increases the collective cultural and touristic offerings of cities, thus contributing to a decrease in tourist pressure on historic city centers. A demonstration is made of how industrial heritage generates alternative tourist spaces according to the principles of cultural sustainability. To verify this hypothesis, the main objective of this work was to determine the operability of old industrial facilities currently converted into cultural spaces; that is, to diagnose what aspects are favoring or hindering the tourism value of the industrial heritage under study. In order to undertake this, from a methodological perspective, a system of indicators has been designed and implemented that, when instrumentalized into a matrix, allow for the quantification of the cultural representativeness and tourist operation of cultural spaces articulated in antique industrial infrastructures.

For the design of the indicators, those few prior investigations focused on industrial heritage that employ this methodology have been taken as reference [56–58]. This limited bibliography has been complemented by a review of similar publications focused on other modalities of cultural heritage [59–61] and its urban management [19,62,63]. The indicators designed here have been replicated in other studies, thus allowing their adjustment for functionality [55,64]. As shown in Table 1, a total of 15 indicators have been designed, structured into five main sections ("Heritage operation", "Cultural operation", "Urban landscape", "Tourist operation", and "Destination construction"). The first three sections are integrated into Group 1, consisting of indicators related to the "Representativeness of cultural spaces of industrial origin"; the last two sections form Group 2, "Tourist use of cultural spaces of industrial origin". Through the first group, we could quantify the exceptionality of the cultural values possessed by the old industrial infrastructures observed; through the second group, we could assess their integration into the tourist dynamics of the three cities. Once the initial results were obtained, they were further classified into four levels: "very high", "high", "medium", and "very low", depending on the quartiles of each city, in such a way that all the cultural spaces in the cities adhere to one of these values. The quantification of each item has been articulated on a 0-to-3 Likert scale, which permits the information to be statistically analyzed. The evaluation criteria of this scale have been debated, agreed upon, and rated by the researchers, and are listed in Table 2.

Once the items were quantified, the results were transferred to a matrix of "Cultural Representativeness and Tourist Use", generated by means of a scatter plot. The indicators from Group 1 were taken as a reference to measure "cultural representativeness", and the indicators from Group 2 were used to evaluate "tourist use". The data corresponding to "cultural representativeness" are located on the *x*-axis of the scatter graph, and those of "tourist use" on the *y*-axis. The vertical and horizontal lines dividing the grid have been established based on the means of the valuations of each group. The matrix has been structured in four quadrants, the definitions of which are given in Table 3.

Finally, from a cartographic point of view, heat maps (density of points) of the three cities have been made, with a triple objective: (i) to locate the cultural spaces of industrial origin, as well as the 30 tourist attractions most highly valued by the users of the tourist portal "Tripadvisor"; (ii) to indicate the position of these two types of elements in relation to their respective historic centers; and (iii) to show the superimposition of spaces with the highest cultural density in relation to industrial heritage and tourist attractions based on the valuations given by tourists. With cartography, the aim was to detect density patterns that allow conclusions to be drawn about the locations of tourist attractions in historical centers and the possibility that cultural spaces of industrial origin are generating alternative tourist areas that help reduce the pressure by tourism on historic centers.

**Table 1.** Indicators of "Representativeness of cultural spaces of industrial origin and tourist use".

| Indicators/Cultural Spaces of Industrial Origin | | | Cultural Space 1 | Cultural Space 2 | Cultural Space *n* | Total |
|---|---|---|---|---|---|---|
| **Group 1: Representativeness of cultural spaces of industrial origin** | Heritage operability | 1. Singularity | | | | |
| | | 2. Historical significance | | | | |
| | | 3. Administrative protection | | | | |
| | Cultural operability | 4. Cultural planning | | | | |
| | | 5. Public use of the space | | | | |
| | | 6. Conservation | | | | |
| | Urban landscape | 7. Improved local environment | | | | |
| | | 8. Integration into the urban landscape | | | | |
| | | 9. Proximity to similar goods | | | | |
| **Group 2: Tourist use of cultural spaces of industrial origin** | Tourist operability | 10. Accessibility | | | | |
| | | 11. Tourist use | | | | |
| | | 12. Tourism planning | | | | |
| | Destination construction | 13. Promotion | | | | |
| | | 14. Commercialization | | | | |
| | | 15. Online positioning | | | | |
| **TOTAL** | | | | | | |

**Table 2.** Criteria for the evaluation of cultural spaces of industrial origin.

| Group | Subgroup | Criteria |
|---|---|---|
| **Group 1: Representativeness of cultural spaces of industrial origin** | Heritage operability | Exceptionality of the cultural value of cultural spaces of industrial origin and levels of administrative protection |
| | Cultural operability | Level of functionality of the space as a cultural resource and state of conservation |
| | Urban landscape | Ability to improve the environment where the cultural space is located |
| **Group 2: Tourist use of cultural spaces of industrial origin** | Tourist use | Level of functionality of the space as a tourist resource |
| | Destination construction | Capacity of the cultural space to become part of the tourism offerings of the city |

**Table 3.** Quadrants of the matrix of "Representativeness of cultural spaces of industrial origin and tourist use". Source: Own elaboration, taking as a reference Cortes and Aranda [61].

| **Group 2 (Quadrant 2): Resources to promote from a tourism perspective** | **Group 1 (Quadrant 1): Strategic resources** |
|---|---|
| High cultural representativeness Low tourist use | High cultural representativeness High tourist use |
| **Group 3 (Quadrant 3): Resources with little differentiation** | **Group 4 (Quadrant 4): Resources with tourism potential** |
| Low cultural representativeness Low tourist use | Low cultural representativeness High tourist use |

This quantitative methodology has been complemented with a bibliographic review related to the history and industrial heritage of the cities selected for the case study.

Books, book chapters, and journal articles indexed in impact databases, such as WoS or SCOPUS, have been consulted, along with technical documents from official institutions and bodies. Statistical sources have proven key to this research, especially those related to aspects of tourism. In addition to national statistical institutes, TourMIS (a project of the Austrian National Tourist Office that collects tourism data from European cities) has been consulted [65]. Fieldwork was carried out in the three European capitals (Madrid, Copenhagen, and Brussels) between 2017 and 2019, making it possible to observe the dynamics of touristic enhancement of industrial heritage in situ and to obtain representative graphic materials. Finally, the data obtained allowed for the generation of graphic material (Tables 3, A1 and A2 and Figures A1 and A2 in the Appendix A), which facilitated the interpretation of the results obtained, as well as the extraction of conclusions.

*3.2. Selection Criteria for Case Studies*

This work methodology has been applied to the three aforementioned European cities to obtain a comparative study that enables the determination of behavior patterns. The selection of these three cases meets various criteria. In the first place, it was considered convenient to analyze the European region where the processes of urban deindustrialization have exhibited common patterns (albeit with many nuances); therefore, the comparison is relatively homogeneous. Second, capital cities were selected given the relative ease with which relevant data can be obtained from similar statistical sources, favoring the validity of the analysis. Finally, a geographic component has also been present, examining cities considered medium-sized on a world scale and distributed into three European regions that may provide different casuistry: Copenhagen for the Northern region, Brussels for the Central region, and Madrid for the Southern region.

A study of this type could not be undertaken without cities possessing considerable industrial heritage, or without scientific studies offering the necessary data with which to correctly assess the designed indicators. Copenhagen, Brussels, and Madrid all presented adequate sources of information to facilitate this comparative study. In 2007, the Danish Agency for Culture assembled a list of 25 cultural assets of industrial origin in order to delve into the economic history of the country to protect this typology of cultural heritage (declaring them "Industrial Sites of National Interest") and to configure new resources susceptible to cultural and tourist use [66]. The Danish Parliament provided funds to cultural institutions to meet this goal, but the 2008 economic crisis negatively affected the development of this project. A total of 10 of the properties that were declared "Industrial Sites of National Interest" are located in Copenhagen, and half of these are considered cultural spaces [67]. In Belgium, the government of the Brussels Capital Region heads the "Inventory of Architectural Heritage of Brussels"—a project launched in 1975 and expanded over the decades to include specific thematic inventories that can be consulted online [68]. This vast catalog contains more than 40,000 locations in the Belgian capital, with a section dedicated exclusively to industrial heritage, involving 827 assets and 234 sites located within the Brussels Capital Region. For this research, eight protected cultural spaces of industrial origin were selected, to which the Tour et Taxis and the Kanal Centre Pompidou were added due to the cultural relevance of their current uses. Finally, in Madrid, 15 industrial facilities have been declared "Assets of Cultural Interest"—the maximum protection granted by the Spanish State to a cultural element. In addition, the Madrid City Council has cataloged seven other elements, granting them some type of protection [55]. For this research, 17 cultural spaces of industrial origin that enjoy certain administrative protection have been taken as reference, along with two other spaces that, while are without such legal status, are nonetheless part of the city's cultural and tourism offerings.

In addition to the above, these three cities selected as case studies are by no means alien to the starting point of this research: the unprecedented growth experienced in urban tourism until 2020. Three types of data show that Copenhagen, Brussels, and Madrid have all been subject to significant increases in tourist flows: the evolution of overnight stays by tourists (in the period of 2014–2019); the increase in tourist intensity (in the period of

2014–2019); and the locations of the main tourist attractions in the historic centers (the year of 2020).

Table 4 presents the evolution in overnight stays by national and international tourists that occurred during the 2014–2019 period. Overnight stays increased across practically the entire time series, reflecting very significant percentages in some cases. For example, Brussels, in 2017, experienced a growth of 13.7%; Copenhagen, in 2018, saw an increase of 8%; and Madrid, in 2017, recorded a 6.6% rise. These data confirm the results obtained that are related to tourist intensity, presented in Table 5. In the three selected cities, a parameter has been applied, relating the number of overnight stays by national and international tourists to the total resident population in the three capitals, and thus generating the "Tourist Intensity Indicator" (TII). This indicator was developed in reference to those tested and validated by TourMIS [65], as well as those of Ditcher and Guevara [69] and Hidalgo et al. [70]. The TII allows for the assessment of tourism intensification for the six years prior to 2020 and, as shown in Table 5, this increased in all three cities during almost the entire series analyzed here.

**Table 4.** Overnight stays (total foreign and domestic) in all forms of paid accommodation, in city areas only. Source: TourMIS [65] and INE [71].

| Year | Copenhagen | % | Brussels Region | % | Madrid | % |
|------|-----------|-----|-----------------|-------|------------|-----|
| 2019 | 9,617,978 | 6.7 | 7,426,142 | 6.2 | 20,850,283 | 5.9 |
| 2018 | 9,014,627 | 9.0 | 6,993,061 | 10.9 | 19,684,880 | 1.8 |
| 2017 | 8,266,908 | 4.3 | 6,306,369 | 13.7 | 19,331,895 | 6.6 |
| 2016 | 7,927,330 | 4.6 | 5,547,772 | −18.3 | 18,138,278 | 1.8 |
| 2015 | 7,581,490 | 9.3 | 6,789,083 | 2.7 | 17,818,431 | 7.9 |
| 2014 | 6,937,121 | 6.7 | 6,611,933 | 6.2 | 16,520,205 | 5.9 |

**Table 5.** Tourist Intensity Indicator (TII). Source: TourMIS [65] and INE [71].

| Year | Copenhagen | Brussels Region | Madrid |
|------|-----------|-----------------|--------|
| 2019 | 15.4 | 6.1 | 6.4 |
| 2018 | 14.7 | 5.8 | 6.1 |
| 2017 | 13.7 | 5.3 | 6.1 |
| 2016 | 13.4 | 4.7 | 5.7 |
| 2015 | 13.1 | 5.8 | 5.7 |
| 2014 | 12.2 | 5.7 | 5.2 |

Regarding the locations of tourist attractions most highly valued by the users of the "Tripadvisor" platform [72], Figures 2–4 show a high concentration of these in the historic centers of all three case studies. In Copenhagen, for example, 66.6% of these resources are located in its historic center (Figure 2); in Brussels, this proportion is 43.3% (Figure 3); and in Madrid, it is 46.6% (Figure 4). These high concentrations of tourist attractions into very specific areas increase the touristic pressure on historic centers, thus aggravating the effects of tourism and compromising the sustainability of the destinations. In contrast, the locations of the cultural spaces of industrial origin are mostly on the peripheries of the historic centers. The objective of this research was precisely to determine whether this sort of cultural attraction (already integrated into a given city's cultural offerings) is assisting in the generation of a second area of tourism influence that might both increase and diversify the supply of attractions and reduce the concentration of flows, thereby favoring cultural and touristic sustainability.

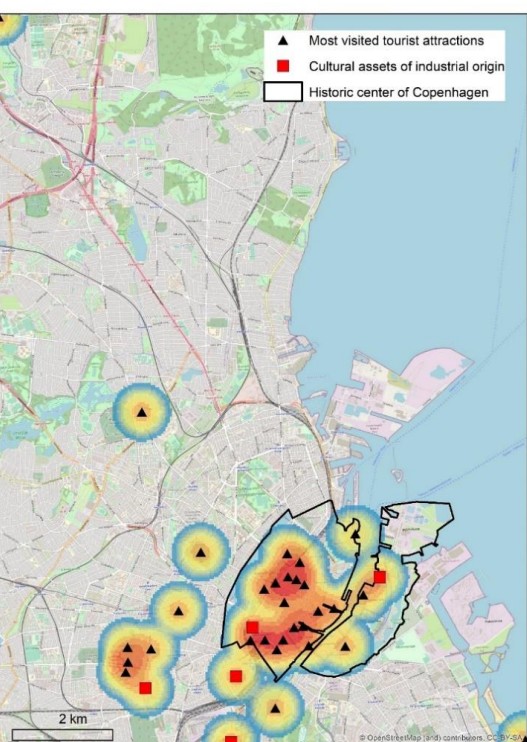

**Figure 2.** Locations of the 30 tourist attractions in Copenhagen most highly rated by Tripadvisor users, and locations of cultural spaces of industrial origin. Source: Own elaboration, taking Tripadvisor (2020) and fieldwork as references.

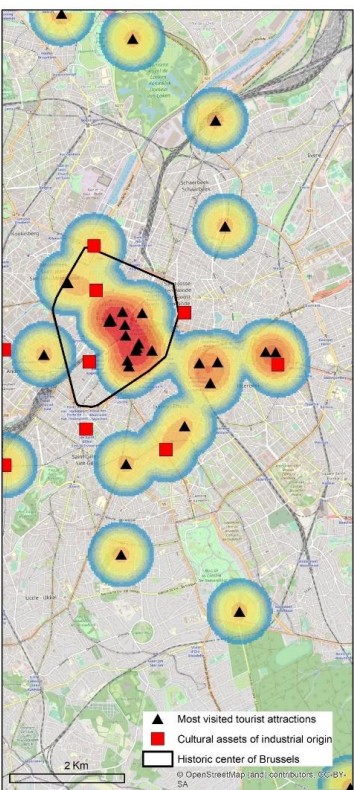

**Figure 3.** Locations of the 30 tourist attractions in Brussels most highly rated by Tripadvisor users, and locations of cultural spaces of industrial origin. Source: Own elaboration, taking Tripadvisor (2020) and fieldwork as references.

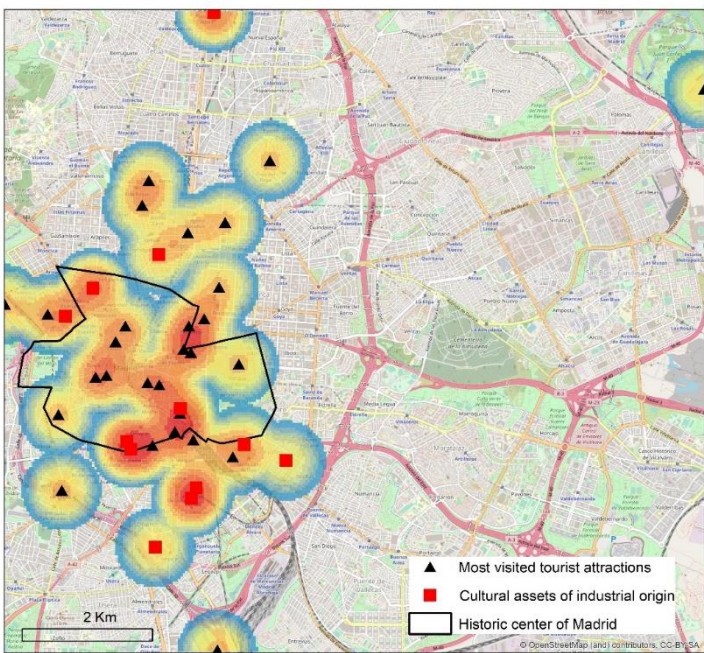

**Figure 4.** Locations of the 30 tourist attractions in Madrid most highly rated by Tripadvisor users, and locations of cultural spaces of industrial origin. Source: Own elaboration, taking Tripadvisor (2020) and fieldwork as references.

## 4. Case Studies: Copenhagen, Brussels, and Madrid

*4.1. A Brief Characterization of the Industrial Heritage in the Selected Cases*

There are currently five cultural spaces of industrial origin in Copenhagen considered to be of national interest. Of these, two cultural assets are representative examples of the agri-food sector: the Carlsberg Brewery and the old Municipal Slaughterhouse. The first, founded in 1848 by J.C. Jacobsen, is located on a small hill between the districts of Valby and Vesterbro, south of the city. Following the transfer of production to a new site in 2008, the factory was reconfigured as a museum that interprets the processes of beer production and the history of the company, and the Carlsberg Brewery currently remains integrated as a cultural resource of industrial origin in the city's tourist offering [73]. For its part, the old Municipal Slaughterhouse, located in the Vesterbro neighborhood very near the city's historic center, has been more affected by the processes of urban gentrification. After cessation of the site's productive activity, the Copenhagen City Council developed a master plan with the aim of transforming it into a "cluster" where traditional companies coexist with exhibition spaces, restaurants, and creative enterprises [66].

In the category of urban infrastructures, the Vandvaerk Pumpethuset stands out—a treatment plant and water distributor in the city that operated between 1859 and 1951. Today it serves as a concert hall, and a portion of the facilities are occupied by a kindergarten. The H.C. Orsted facility, construction of which began in 1916, remains operational, with the main function of providing heating and hot water to homes and commercial establishments in the city. It is remarkable for its architecture, featuring red brick and blue painted chimneys. Within this power plant, it has been possible since 2006 to visit the DieselHouse—an interactive exhibition that revolves around different diesel engines, especially the Burmeister & Wain engine that, for over 30 years, was among the largest in the world [74]. Finally, the closure in the 1990s of the city's Holmen shipyards and naval base prompted the urban transformation of an area characterized by the mixed presence of residential, business, educational, and residual military buildings. The installation of the Opera House in 2005 culminated this process of reform and marked its opening to society. The traditional uses of these cultural spaces of industrial origin in Brussels are multiple and varied, in concordance with the heterogeneity of industry in that city. Warehouses and

deposits such as the Anciens Grands Magasins Merchie-Pède and the Palais du Vin Brias et Cie predominate, along with small artisan workshops like the Ancienne Maison Hoguet (currently the Maison du Livre, or Book House), the Anciens Établissements Mommen, and the Imprimerie Agefi. Other elements are linked to the food and beverage industry, such as the "Marché aux bestiaux couvert de Cureghem" slaughterhouse; or the Wielemans-Ceuppens brewery. Additionally, noteworthy are the Autoworld exhibition in the Parc du Cinquantenaire (situated within a hall of iron, glass, and stone, and built to house an 1880 exhibition celebrating the 50th anniversary of Belgian independence) and the Casa Solvay, designed by architect Víctor Horta for the Solvay family of industrial entrepreneurs.

Most of the industrial sites analyzed have undergone changes in use. For example, the old Wielemans-Ceuppens brewery has been transformed into the Wiels contemporary art center, and the Hoguet mansion is now the Book House, featuring literary activities. The Merchie-Pède warehouses and the Brias et Cie Wine Palace, both located on Rue des Tanneurs, now offer coworking spaces, workshops, retail shops, accommodations, and restaurants [75] Finally, the Imprimerie Agefi print facility has been transformed into Madame Moustache, a concert hall. Additionally worthy of mention are those former industrial sites that have undergone smaller modifications from their traditional uses: the antique "Marché aux bestiaux couvert de Cureghem" slaughterhouse is now known simply as Abattoir, a market selling food products as well as offering cultural and educational activities; the Établissements Mommen, originally a workshop for artisans and painters, is now an exhibition center that promotes the work of young artists; the Tour et Taxis, once a warehouse complex, now serves as a multifunctional space with shops, restaurants, and offices; and the Garage Citröen, constructed as an automobile service station, now houses the new headquarters of the Pompidou Center in Brussels.

During the first half of the 20th century, Madrid positioned itself as an industrial pole of national reference within the autarkic policies of Francoism. However, beginning in the 1970s, there was the relocation of industrial facilities on the outskirts of the city [76]. This generated a paradoxical situation: on the one hand, the destruction of the production equipment that had been disaffected; on the other, protection and conservation by public administrations of certain industrial buildings, some of which are regarded as cultural heritage today. As they were in Brussels, Madrid's industrial infrastructures were addressed to varied uses. Within the agri-food sector, manufacturers of note included the Pacisa biscuit factory (currently the Circo Price), the El Aguila and Mahou breweries (today converted into the Joaquín Leguina Library and the ABC Museum, respectively), and a former tobacco factory that now houses two separate cultural spaces, the CSA La Tabacalera de Lavapiés and Tabacalera Art Promotion. The food sector also included the municipal slaughterhouse, now a center for contemporary culture known as Matadero Madrid.

Further of interest is the San Miguel Market, which (although sustaining its original use) has undergone a process of touristification oriented to a gourmet public. The city's water supply, configured by a network of reservoirs, currently provides sites for two exhibition spaces, the Canal Foundation and the Isabel II Exhibition Hall. Elsewhere, further development of the railroad and suburban railway systems forced the decommission of transport stations that were converted into the Railway Museum of Madrid and the Atocha Station Tropical Garden. Old transport workshops have been rehabilitated as art centers, such as La Neomudéjar, and a disused metro station was later reopened to the public as Chamberí Station/Andén 0 (Platform 0), maintaining its original appearance. The introduction of electric power drove the opening of many power plants, two of which were conserved after obsolescence and transformed into cultural spaces: Motor Warehouse/Andén 0 and the Caixa Forum. Finally, there is the case of Medialab Prado, a former sawmill that is today a center for digital culture.

### 4.2. The Case of Copenhagen (Denmark)

With the aim of evaluating the current tourist operation of these cultural spaces of industrial origin, the methodology detailed in the above Materials and Methods section

was applied. First, each of the cultural assets was evaluated in relation to the indicators of cultural representativeness and tourism use. As shown in Table 6, one of the sites (the Carslberg Brewery) scored "very high", with 38 points; two cultural spaces (the Vandvaerk Punpethuset and Meatpacking District) obtained a "high" value in tourist operation, with 36 points each; and the Holmen Royal Docks and Orsted Power Station ranked "low" and "very low" in terms of value, at 32 and 20 points, respectively. Subsequently, the data obtained was applied to the matrix of "Cultural Representativeness and Tourist Use", collected in Figure 5.

**Table 6.** Matrix of "Representativeness of cultural spaces of industrial origin and tourist use". Cultural spaces of industrial origin in Copenhagen. Source: Own elaboration.

| Indicators/Cultural Spaces of Industrial Origin | | | Carlsberg Brewery | Pumpethuset | Meatpacking District | Holmen Royal Docks | H. C. Ørsted Power Station | Total |
|---|---|---|---|---|---|---|---|---|
| **Group 1: Representativeness of cultural spaces of industrial origin** | Heritage operability | Singularity | 2 | 1 | 2 | 2 | 2 | **11** |
| | | Historical significance | 3 | 1 | 3 | 3 | 2 | **14** |
| | | Administrative protection | 1 | 1 | 1 | 1 | 1 | **6** |
| | Cultural operability | Cultural planning | 3 | 2 | 3 | 3 | 1 | **13** |
| | | Public use of the space | 2 | 2 | 3 | 2 | 1 | **11** |
| | | Conservation | 3 | 3 | 3 | 3 | 3 | **17** |
| | Urban landscape | Improvement of local environment | 3 | 3 | 3 | 3 | 0 | **15** |
| | | Integration into the urban landscape | 2 | 3 | 3 | 3 | 0 | **14** |
| | | Proximity to similar goods | 3 | 3 | 3 | 3 | 3 | **18** |
| **Group 2: Tourist use of cultural spaces of industrial origin** | Tourism operability | Accessibility | 2 | 3 | 3 | 2 | 2 | **14** |
| | | Tourist use | 3 | 3 | 3 | 2 | 2 | **15** |
| | | Tourism planning | 3 | 3 | 3 | 3 | 1 | **15** |
| | Destination construction | Promotion | 2 | 3 | 3 | 1 | 1 | **11** |
| | | Commercialization | 3 | 3 | 0 | 0 | 1 | **7** |
| | | Online positioning | 3 | 2 | 0 | 1 | 0 | **6** |
| **TOTAL** | | | **38** | **36** | **36** | **32** | **20** | |



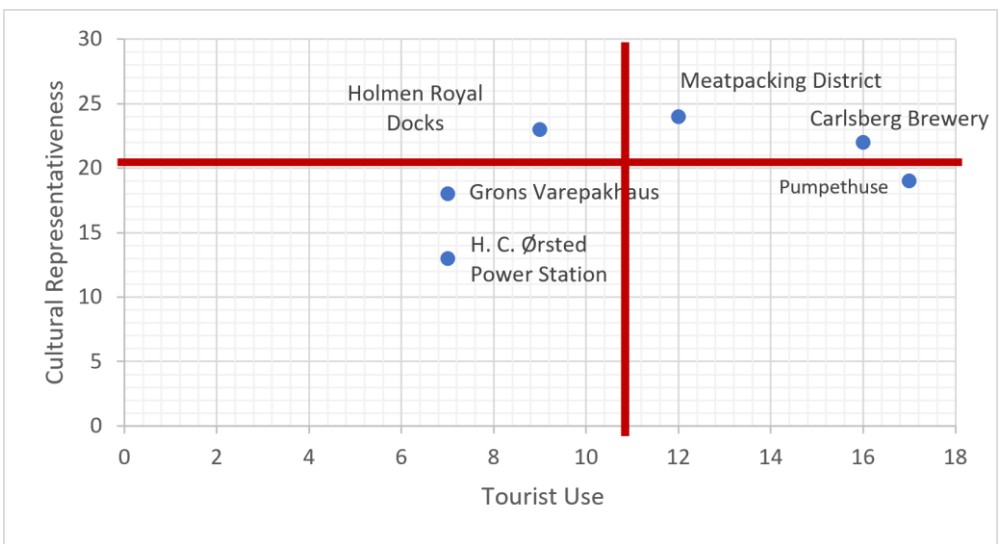

**Figure 5.** Matrix of "Representativeness of cultural spaces of industrial origin and tourist use" in Copenhagen. Source: Own elaboration.

Only one of these strategic resources, the Carlsberg Brewery, is characterized by both high tourist use and high cultural representativeness. In contrast, the Orsted Power Station is characterized as a cultural space of industrial origin with low tourist use and low cultural representativeness ("Resources with little differentiation"). On the other hand, both the Holmen Royal Docks and Meatpacking District reveal low tourist use but high cultural representativeness ("Resources to promote from a tourist perspective"), while the Pumpethuset shows high tourist use but low cultural representativeness ("Resources with tourist potential").

Following the evaluation of the results obtained, it can be concluded that the cultural spaces of industrial origin in Copenhagen exhibit a series of positive aspects that favor their tourism value. In the first place, the declaration of these as "Industrial Sites of Interest" shows the interest of public administrations in this type of heritage, although it should be noted that no other administrative protections have been detected. On the other hand, the cultural representativeness of all these assets scored high, along with the state of conservation and the proximity between them. From the point of view of touristic operation, tourism planning and use are detected in several, particularly the highly touristed Carlsberg Brewery. Regarding those aspects that hinder touristic operation, those related to destination construction are indicated as weak tourism promotion and marketing or limited online positioning. Indeed, with the exception of the Carlsberg Brewery there are no complex tourist products based on the industrial heritage of Copenhagen that serve to explain production processes of the past, nor has the implementation of a tourism modality focused on the knowledge of industry been detected.

*4.3. The Case of Brussels (Belgium)*

The assessment of the 10 selected cultural spaces of industrial origin in Brussels (Table 7) indicates that Autoworld achieved "very high" cultural representativeness as well as tourist use. There were two sites (Kanal Centre Pompidou and Tour et Taxis) that scored "high", four others (Abbatoir, Solvay House, Wiels Art Center, Ateliers des Tanneurs) scored "low", and three (Madame Moustache, Ateliers Mommen, Book House) scored "very low". In regard to the matrix of "Cultural Representativeness and Tourist Use" (Figure 6), Autoworld is considered a strategic resource, developing both "high" cultural representativeness and tourist use. On the other hand, Atelieres Mommen, Book House, Wiels Art Center, Solvay House, Madame Moustache, Ateliers des Tanneurs, and Kanal Centre Pompidou are all in the reverse position, considered resources of little differentiation that scored "low" in terms of both cultural representativeness and tourist operation. Finally,

the Abbatoir and Tour et Taxis rated "high" in cultural representativeness but "low" in tourist activity, classifying them as resources to be promoted from a tourist perspective. No resources were found to have relevant tourism potential.

**Table 7.** Matrix of "Representativeness of cultural spaces of industrial origin and tourist use". Cultural spaces of industrial origin in Brussels. Source: Own elaboration.

| Indicators/Cultural Spaces of Industrial Origin | | | Autoworld | Kanal Centre Pompidou | Tour et Taxis | Abbatoir | Solvay House | Wiels Art Center | Ateliers Des Tanneurs | Madame Moustache | Ateliers Mommen | Book House | Total |
|---|---|---|---|---|---|---|---|---|---|---|---|---|---|
| Group 1: Representativeness of cultural spaces of industrial origin | Heritage operability | Singularity | 3 | 2 | 3 | 3 | 2 | 2 | 2 | 1 | 2 | 2 | **22** |
| | | Historical significance | 3 | 2 | 3 | 2 | 1 | 2 | 2 | 1 | 1 | 1 | **18** |
| | | Administrative protection | 2 | 1 | 1 | 2 | 2 | 2 | 2 | 2 | 2 | 2 | **18** |
| | Cultural operability | Cultural planning | 3 | 3 | 2 | 3 | 3 | 3 | 2 | 2 | 2 | 3 | **26** |
| | | Public use of the space | 3 | 3 | 2 | 3 | 1 | 3 | 2 | 1 | 2 | 1 | **21** |
| | | Conservation | 3 | 3 | 3 | 3 | 3 | 3 | 3 | 3 | 3 | 3 | **30** |
| | Urban landscape | Improvement of local environment | 3 | 3 | 3 | 3 | 3 | 3 | 3 | 2 | 2 | 2 | **27** |
| | | Integration into the urban landscape | 3 | 3 | 3 | 3 | 3 | 3 | 3 | 2 | 2 | 2 | **27** |
| | | Proximity to similar goods | 2 | 2 | 2 | 1 | 2 | 1 | 3 | 3 | 2 | 2 | **20** |
| Group 2: Tourist use of cultural spaces of industrial origin | Tourism operability | Accessibility | 3 | 3 | 3 | 2 | 2 | 2 | 2 | 2 | 2 | 2 | **23** |
| | | Tourist use | 3 | 3 | 2 | 1 | 1 | 2 | 1 | 2 | 1 | 1 | **17** |
| | | Tourism planning | 3 | 3 | 2 | 1 | 1 | 1 | 1 | 1 | 1 | 1 | **15** |
| | Destination construction | Promotion | 3 | 2 | 2 | 1 | 3 | 1 | 2 | 1 | 1 | 1 | **17** |
| | | Commercialization | 3 | 3 | 2 | 1 | 2 | 0 | 0 | 3 | 0 | 0 | **14** |
| | | Online positioning | 3 | 3 | 3 | 3 | 2 | 3 | 2 | 3 | 2 | 2 | **26** |
| **TOTAL** | | | **43** | **39** | **36** | **32** | **31** | **31** | **30** | **29** | **25** | **25** | |

Taking into account the results derived from the application of the indicators and the matrix of "Cultural Representativeness and Tourism Use" (Figure 6), it can be said that online information, accessibility, and the integration of these cultural assets of industrial origin into the urban landscape of the city are three factors that most favor their inclusion in the city's cultural offerings and tourist promotion. In contrast, the absence of tourism planning, whether in strategic terms or in promotional actions by public and private agents, are two factors that hamper their efforts for touristic revitalization. Indeed, the scarcity of tourist products based on the city's industrial heritage proves especially significant; practically no tourist brochures or online portals make reference to the city's heritage, with the notable exception of La Fonderie, an NGO that studies the industrial heritage of Brussels and offers guided tours of old industrial spaces [77]. As in Copenhagen, no complex tourist products themed around industrial heritage nor industrial tourism flows to the city were detected.

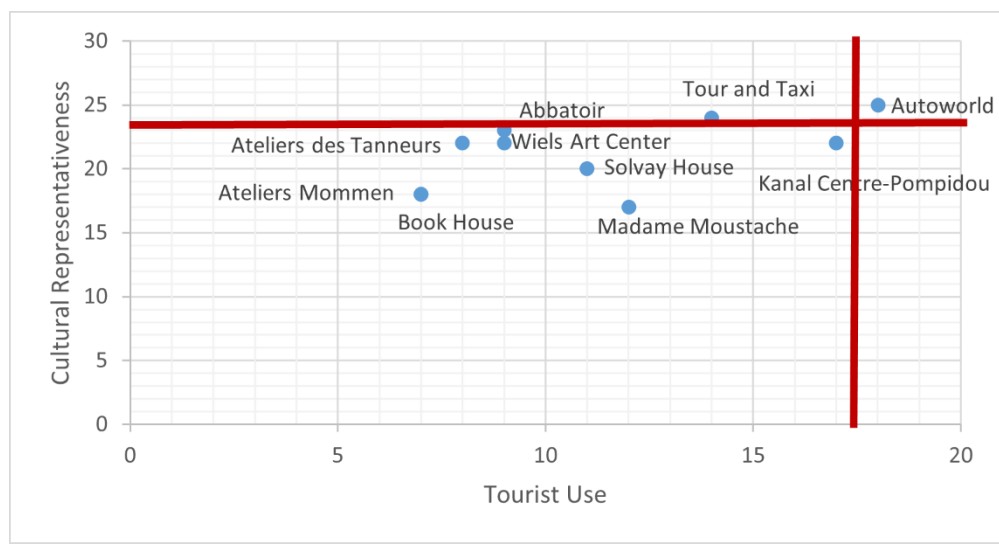

**Figure 6.** Matrix of "Representativeness of cultural spaces of industrial origin and tourist use" in Brussels. Source: Own elaboration.

*4.4. The Case of Madrid (Spain)*

When these 17 cultural spaces of industrial origin in Madrid were evaluated (Table 8), nine of them scored "very high" in terms of their "Cultural Representativeness and Tourist Use" (San Miguel Market, Matadero Madrid, Tabacalera Art Promotion, Caixa Forum, Royal Tapestry Factory, Canal Foundation, Madrid Railway Museum, Joaquín Leguina Regional Library, and CSA La Tabacalera de Lavapiés). Another site, the ABC Museum of Drawing and Illustration, is considered "high", while La Neomudéjar and Medialab Prado rated "low", and the remaining five (Canal of Isabel II Exhibition Center, Atocha Station Tropical Garden, Circo Price, Motor Warehouse/Andén 0, Chamberí Station/Andén 0) rated "very low".

Regarding the matrix of "Cultural Representativeness and Tourist Use" (Figure 7), five cultural spaces are positioned as strategic resources with "high" cultural representativeness as well as "high" tourist use (Tabacalera Art Promotion, Matadero Madrid, San Miguel Market, Caixa Forum, Canal Foundation). In contrast, six cultural spaces scored "low" in terms of both cultural representativeness and tourist use (Atocha Station Tropical Garden, Medialab Prado, Isabel II Exhibition Hall, Circo Price, Motor Warehouse/Andén 0, Chamberí Station/Andén 0) and are considered to be resources with little differentiation. Two cultural spaces, the ABC Museum and La Neomudéjar, showed "low" cultural representativeness but "high" tourist activity and are therefore considered resources with strong potential. Finally, the Royal Tapestry Factory, the Railway Museum of Madrid, the CSA La Tabacalera de Lavapiés, and the Joaquín Leguina Library all ranked "high" in terms of cultural representativeness but "low" in tourist activity. These four cultural spaces of industrial origin are therefore considered resources to be promoted from a tourist perspective.

**Table 8.** Matrix of "Representativeness of cultural spaces of industrial origin and tourist use". Cultural spaces of industrial origin in Madrid. Source: Own elaboration.

| Indicators/Cultural Spaces of Industrial Origin | | | San Miguel Market | Matadero Madrid | Tabacalera Art Promotion | Caixa Forum | Royal Tapestry Factory | Canal Foundation | Railway Museum of Madrid | Joaquín Leguina Library | CSA La Tabacalera of Lavapiés | ABC Museum of Drawing and Illustration | Medialab Prado | La Neomudéjar | Isabel II Exhibition Hall | Atocha Station Tropical Garden | Circo Price | Motor Warehouse/Andén 0 | Chamberí Station/Andén 0 | Total |
|---|---|---|---|---|---|---|---|---|---|---|---|---|---|---|---|---|---|---|---|---|
| **Group 1: Representativeness of cultural spaces of industrial origin** | Heritage operability | Singularity | 3 | 3 | 3 | 1 | 3 | 2 | 3 | 2 | 3 | 1 | 1 | 1 | 2 | 1 | 1 | 1 | 1 | **32** |
| | | Historical significance | 2 | 3 | 3 | 1 | 3 | 2 | 3 | 2 | 3 | 1 | 1 | 1 | 1 | 1 | 1 | 1 | 1 | **30** |
| | | Administrative protection | 3 | 2 | 3 | 2 | 3 | 3 | 3 | 3 | 3 | 1 | 1 | 1 | 1 | 2 | 2 | 1 | 1 | **35** |
| | Cultural operability | Cultural planning | 3 | 3 | 3 | 3 | 3 | 3 | 2 | 2 | 3 | 2 | 2 | 3 | 1 | 1 | 2 | 1 | 1 | **38** |
| | | Public use of the space | 2 | 3 | 3 | 3 | 3 | 3 | 3 | 3 | 3 | 3 | 2 | 2 | 2 | 3 | 1 | 1 | 1 | **41** |
| | | Conservation | 3 | 3 | 3 | 3 | 3 | 3 | 3 | 3 | 1 | 3 | 3 | 2 | 3 | 3 | 1 | 3 | 3 | **46** |
| | Urban landscape | Improvement of local environment | 3 | 3 | 3 | 3 | 3 | 3 | 3 | 3 | 3 | 3 | 3 | 1 | 3 | 3 | 2 | 2 | 1 | **45** |
| | | Integration into the urban landscape | 3 | 3 | 3 | 3 | 3 | 3 | 2 | 3 | 3 | 3 | 3 | 1 | 3 | 3 | 1 | 2 | 1 | **43** |
| | | Proximity to similar goods | 2 | 2 | 3 | 3 | 3 | 1 | 3 | 3 | 3 | 1 | 3 | 3 | 1 | 2 | 3 | 1 | 1 | **38** |
| **Group 2: Tourist use of cultural spaces of industrial origin** | Tourism operability | Accessibility | 3 | 3 | 3 | 3 | 3 | 2 | 3 | 3 | 3 | 3 | 3 | 3 | 3 | 3 | 3 | 3 | 3 | **50** |
| | | Tourist use | 3 | 2 | 2 | 3 | 2 | 2 | 2 | 1 | 1 | 2 | 1 | 1 | 1 | 1 | 2 | 1 | 1 | **28** |
| | | Tourism planning | 3 | 3 | 2 | 3 | 2 | 2 | 2 | 1 | 1 | 2 | 1 | 1 | 1 | 1 | 1 | 1 | 1 | **28** |
| | Destination construction | Promotion | 3 | 3 | 2 | 3 | 2 | 3 | 2 | 2 | 1 | 2 | 2 | 3 | 2 | 1 | 1 | 2 | 2 | **36** |
| | | Commercialization | 2 | 2 | 2 | 3 | 1 | 3 | 1 | 1 | 1 | 2 | 1 | 3 | 2 | 1 | 2 | 2 | 2 | **31** |
| | | Online positioning | 3 | 3 | 2 | 3 | 2 | 3 | 2 | 2 | 1 | 2 | 3 | 3 | 2 | 1 | 3 | 2 | 2 | **39** |
| **TOTAL** | | | **41** | **41** | **40** | **40** | **39** | **38** | **37** | **34** | **33** | **31** | **30** | **29** | **28** | **27** | **26** | **24** | **22** | |

In the case of Madrid, the factors favoring the touristic use of cultural spaces of industrial origin are related to the urban landscape (the degree and quality of integration of the attractions into the urban environment), as well as cultural operability, cultural planning, public use of the space, and state of conservation. Regarding the aspects that hinder the tourist enhancement of these spaces, tourism operations are relevant, especially in their degrees of integration into tourism planning, their use as tourism resources, and their commercialization. Much like in Copenhagen and Brussels, but with the exceptions of the transport-related Motor Warehouse, Railway Museum, and Chamberí Station, there are no complex tourist products in Madrid specialized in publicizing the productive past of the city, and specific industrial tourism has not been detected.

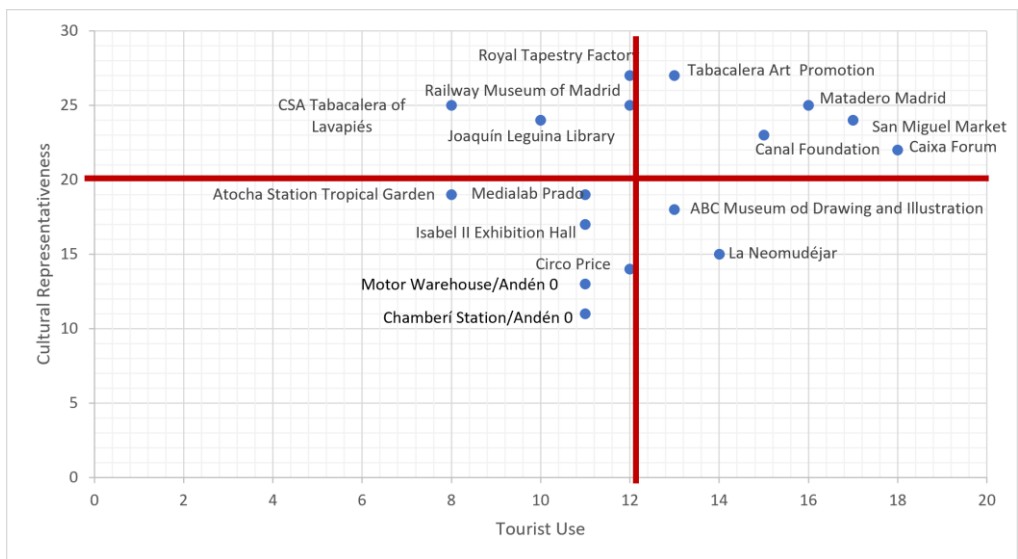

**Figure 7.** Matrix of "Representativeness of cultural spaces of industrial origin and tourist use" in Madrid. Source: Own elaboration.

## 5. Results Obtained

The matrix for "Representativeness of cultural spaces of industrial origin and tourist use" has permitted the classification of the 32 cultural spaces of industrial origin under study into four large groups: strategic resources; resources to promote from a tourist perspective; resources with little representation; and, finally, resources with tourism potential (Table 9).

**Table 9.** Classification of cultural spaces of industrial origin based on the "Matrix of representativeness of cultural spaces of industrial origin and tourist use". Source: Own elaboration.

| Group 2: Resources to promote from a tourism perspective High cultural representativeness Low tourist use | Group 1: Strategic resources High cultural representativeness High tourist use |
|---|---|
| Royal Tapestry Factory Railway Museum of Madrid Joaquín Leguina Library CSA La Tabacalera de Lavapiés Tour et Taxis Abbatoir Meatpacking District Holmen Royal Docks | Autoworld Carlsberg Brewery Tabacalera Art Promotion Matadero Madrid San Miguel Market Caixa Forum Canal Foundation |
| **Group 3: Resources with little differentiation** Low cultural representativeness Low tourist use | **Group 4: Resources with tourism potential** Low cultural representativeness High tourist use |
| Tropical Garden Atocha Station Medialab Prado Isabel II Exhibition Hall Circo Price Motor Warehouse/Andén 0 Chamberí Station/Andén 0 Wiels Art Center Ateliers des Tanneurs Madame Moustache Ateliers Mommen Book House H. C. Orsted Power Station | ABC Museum La Neomudéjar Pumpethuset |

In the first quadrant of the matrix, "strategic resources", 7 of the 32 assets analyzed (or 21.8%) are integrated; Madrid has the highest percentage of these, at 71.4%, while Copenhagen and Brussels have 14.2% each. These assets are characterized by high qualifications in conservation, cultural planning, and improvement of the local environment, and by low qualifications in administrative protection, singularity, and historical significance in terms of cultural representativeness (Group 1). Taking into account the aspects related to tourist use (Group 2), high qualifiers include online positioning, promotion, tourism planning, and accessibility, while commercialization is qualified as "low".

The second quadrant of the matrix, "resources to promote from a tourist perspective", is characterized by high cultural representativeness and low tourist use. There were 8 cultural spaces of the 32 studied (25%) that are integrated into this group. Madrid again leads this quadrant with 50% of the assets, followed by Copenhagen and Brussels with 25% each. Regarding the indicators related to cultural representativeness (Group 1), improvement of the local environment and integration into the urban landscape scored the highest, while the lowest values were scored by administrative protection, cultural planning, and singularity. For its part, tourism planning shows a higher valuation in relation to touristic operation (Group 2), together with tourist use, promotion, and accessibility; to the contrary, commercialization was among those recording lower scores.

In the third quadrant of the matrix, "resources with little differentiation", characterized by low cultural representativeness and low tourist use (see Figure 1), 12 cultural spaces of industrial origin are integrated, representing 37.5% of the total. In this case, 50% of the analyzed assets in Madrid belong to this category, followed by Brussels with 41.6% and Copenhagen with 8.3%. The Group 1 indicators showing higher valuations are conservation, improvement of the local environment, and integration into the urban landscape, while singularity and historical significance are valued least. In Group 2, among the indicators for tourist use, online positioning ranked "high", while tourism planning ranked "low".

Finally, resources with tourist potential, which scored "low" in cultural representativeness and "high" in tourist use, represent 9.3% of the total. In Madrid, 66.6% of the cultural assets analyzed are included in this group, and 33.4% in Copenhagen. No assets in this category were detected for Brussels. Regarding cultural representativeness, registering lower values were singularity, historical significance, and administrative protection, while (cultural) conservation ranked "high". Regarding the indicators of touristic operation, accessibility, promotion, and commercialization scored the highest, while tourist use and tourism planning registered the lowest scores.

Therefore, the results derived from the matrix of "Cultural Representativeness and Tourist Use" reveal that the cultural values of spaces of industrial origin are the factors contributing the most to the integration of these assets into the tourism offerings of cities (Figure 8). On the other hand, those aspects found to be holding back integration are precisely those related to tourism operation and destination construction. Therefore, actions of tourist valuation of these spaces must be reinforced to favor their integration into the tourist attractions on offer by these cities. Taking into account the fact that the cultural values of these industrial spaces favor their conversion into tourist resources, and the proposal that tourist operation needs reinforcement so as not to become an obstacle, a statistical correlation test has been carried out. The intention in conducting this test was to verify whether those goods that scored "high" in cultural representativeness also did so in tourist use, but the results proved to be negative. In other words, there is no correlation between the two factors (Appendix A, Table A4).

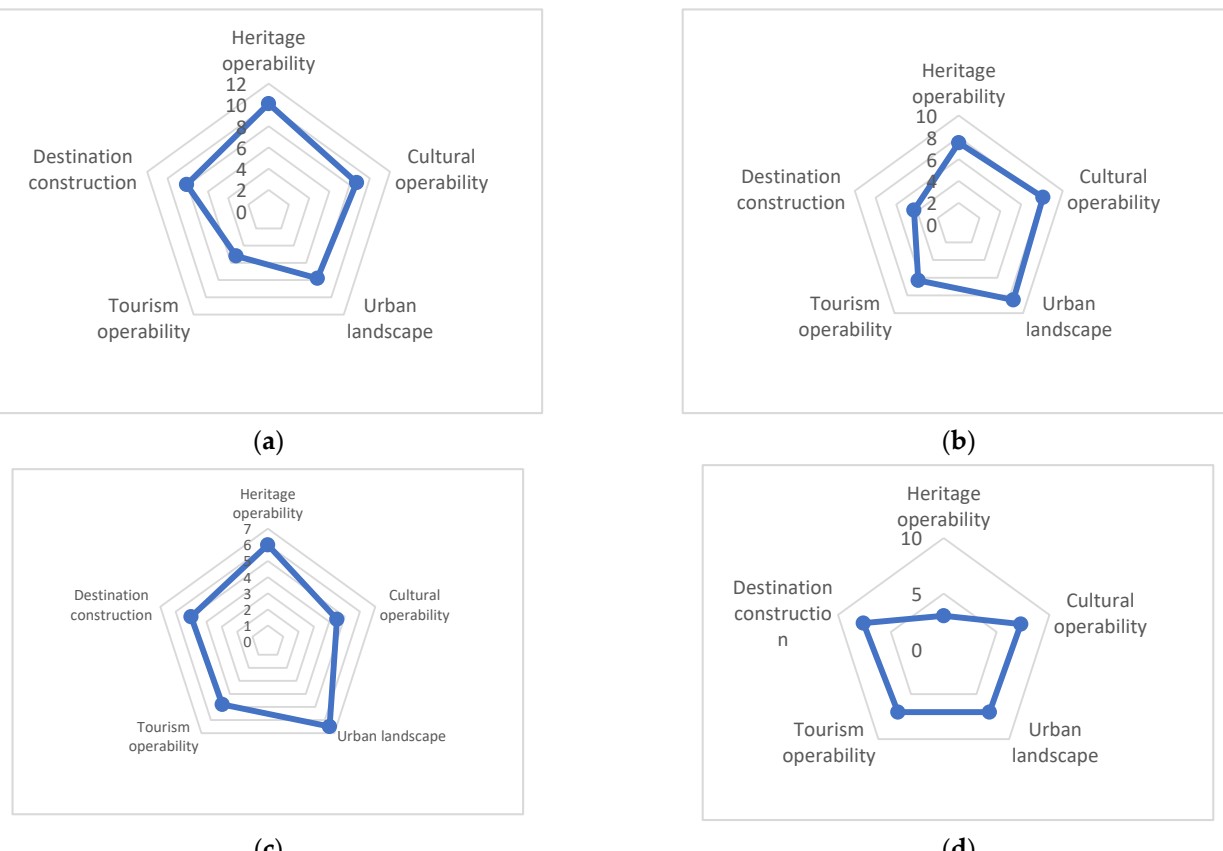

**Figure 8.** Radial graphics corresponding to the classification of cultural spaces of industrial origin, based on the matrix of "Representativeness of cultural spaces of industrial origin and tourist use": (**a**) Strategic resources; (**b**) Resources to promote from a tourist point of view; (**c**) Resources with little differentiation; (**d**) Resources with tourism potential.

As indicated in Section 4.1, the typical location of cultural spaces of industrial origin in areas peripheral to historic centers may ostensibly favor the creation of new tourist spaces. In the case studies analyzed, three representative examples of this particular process were detected: the historic Carlsberg Brewery in Copenhagen, Autoworld in Brussels, and the Matadero cultural complex in Madrid. These three elements are considered strategic resources according to the matrix of "Cultural Representativeness and Tourist Use", possessing high cultural value as well as attracting high tourist interest. All of them are located outside of the historic centers, but as the heat maps show, they have sufficiently significant weight in the overall destination to configure new tourist areas. Thus, it can be said that these spaces are promoting both cultural sustainability (by creating new cultural and tourist resources) and tourism sustainability (by offering alternatives that can reduce the pressure of tourism on historic centers). However—and here is the main weakness detected—there are no tourism products specialized in industrial heritage. As mentioned above, this trend applies to all of the analyzed goods: there is no industrial tourism in any of the three cities under study, despite the existence of resources.

## 6. Discussion

The results obtained allow us to affirm that in the three cities analyzed, not only are cultural spaces of industrial origin already integrated into the tourist destinations on offer (those considered strategic resources), but there are numerous cultural assets in the process of integration. Elements have also been detected that with appropriate actions could conceivably be incorporated into existing urban tourism. Therefore, it can be said that the first part of the working hypothesis proposed in this research—that the transformation of old urban industrial facilities into cultural spaces can increase the cultural and tourist

offerings of the cities—is verified. Industrial heritage does indeed generate tourist spaces alternative to the main tourist attractions in all three of the cities analyzed.

Furthermore, the typical location of these cultural resources on the periphery of the historic centers contributes to reducing the tourist pressure on the latter by offering tourists alternative destinations to those considered traditional. In this way, the second part of the research hypothesis is likewise verified, given that the transformation of old urban industrial facilities into cultural spaces not only increases and diversifies the cultural and tourist offerings of these cities, but also that their generally peripheral locations would favor the reduction of tourist pressure on historic centers by expanding a city's tourism area. The enhancement of industrial heritage thus favors the sustainability of urban tourist destinations by increasing and diversifying the available tourist draws and by fostering concentration.

However, it must be borne in mind that none of these three cities offer complex tourist products specialized in industrial heritage. Instead, old industrial complexes tend to be used as containers, and in most cases, the original industrial uses of the sites are not even explained (with exceptions, including the Carlsberg Brewery and the Railway Museum of Madrid). A form of urban industrial tourism, as seen in the aforementioned mining regions of the Ruhr in Germany, Montceau-les-Mines in France, or in Coal-Brookdale in the United Kingdom, has not been implemented in these cases.

## 7. Conclusions

As a general conclusion, it can be said that this research is aligned with the theoretical proposals expressed by Throsby [42], Nocca [19], and the Basque Observatory of Culture [23], all of which consider that cultural heritage reinforces the cultural sustainability of cities, while also favoring their economic, social, and environmental development. The enhancement of tourism to embrace cultural spaces of industrial origin can serve as an excellent tool with which to address problems derived from the intensification of tourism in cities, also contributing to the sustainability of tourism. However, a process of true valorization of industrial heritage, which culminates in the creation of a destination specialized in industrial tourism, is much more extensive and complex than that observed in the case studies. Although tourist activity linked to industrial tourism has indeed been detected in the cities analyzed, no specialization of urban tourism exists in this industrial modality. Therefore, it can be asserted that an authentic opportunity to continue reinforcing the cultural and tourist sustainability of these cities is being lost, as indicated by Hidalgo, Palacios, and García [78]. Considering this loss of potential, future research must be conducted in order to analyze other phases of the tourism enhancement process and to deepen the understanding of the relationships between industrial tourism and sustainability in urban destinations.

Finally, the COVID-19 pandemic has opened within tourism studies a framework for reflection on the resilience of urban destinations in the face of crisis; the literature analyzing the impact of the pandemic on tourism and cultural activity is already large and varied [79–82]. On the other hand, in regard to the urban destinations where industrial heritage might play a role in the deconcentration of historic centers (such as those analyzed here), no specific bibliography has been detected, suggesting the possibility that future studies could be developed on this topic.

**Author Contributions:** Conceptualization, methodology, formal analysis, and writing—original draft preparation, C.H.-G., A.P.-G. and D.B.-T.; formal analysis and writing—review and editing, C.H.-G., A.P.-G., D.B.-T. and J.A.R.-E.; investigacion fieldtrip, C.H.-G. and A.P.-G.; cartography, J.A.R.-E. and C.H.-G. The supervisory task corresponded to the first signatory author. All authors have read and agreed to the published version of the manuscript.

**Funding:** This research forms part of a competitive project "Culture and Territory in Spain: Processes and Impacts in Small and Medium-Sized Cities" (Ref CSO2017-83603-C2-2-R), financed by the State Research Program "Development and Innovation Oriented to the Challenges of Society" of the Spanish Ministry of Economy and Competitiveness, within the framework of the State Plan for Scientific and Technical Research and Innovation, 2013–2016. The project was developed by the

Research Group in Urban Studies and Tourism (URByTUR) of the Department of Geography of the Autonomous University of Madrid (UAM).

**Institutional Review Board Statement:** Not applicable.

**Informed Consent Statement:** Not applicable.

**Data Availability Statement:** Not applicable.

**Conflicts of Interest:** The authors declare no conflict of interest. The funding organization had no role in the design of the study; in the collection, analysis, or interpretation of data; in the writing of the manuscript; or in the decision to publish the results.

## Appendix A

**Table A1.** Cultural spaces of industrial origin located in Copenhagen.

| N° | Denomination | Traditional Use | Current Use |
|---|---|---|---|
| 1 | Holmen Royal Docks | Shipyards | Multifunctional space |
| 2 | Carlsberg Brewery | Brewery | Interpretation center |
| 3 | Meatpacking District | Municipal slaughterhouse | Multifunctional space |
| 4 | Pumpethuset | Water treatment | Concert hall |
| 5 | H.C. Orsted Power Station | Energy | Interpretation center |

**Table A2.** Cultural spaces of industrial origin located in Brussels.

| N° | Denomination | Traditional Use | Current Use |
|---|---|---|---|
| 1 | Abbatoir | Slaughterhouse | Multifunctional space |
| 2 | Wiels Art Center | Brewery | Art center |
| 3 | Book House | Residence | Literary center |
| 4 | Solvay House | Residence | Tourable monument |
| 5 | Autoworld | Exhibition center | Exhibition center |
| 6 | Ateliers Mommen | Workshops | Exhibition center |
| 7 | Ateliers des Tanneurs | Warehouses | Multifunctional space |
| 8 | Madame Moustache | Print facility | Concert hall |
| 9 | Tour et Taxis | Loading station | Multifunctional space |
| 10 | Kanal-Centre Pompidou | Automotive garage | Art center/other |

**Table 3.** Cultural spaces of industrial origin located in Madrid.

| N° | Denomination | Traditional Use | Current Use |
|---|---|---|---|
| 1 | CSA La Tabacalera de Lavapiés | Tobacco factory | Center for contemporary culture |
| 2 | Tabacalera Art Promotion | Tobacco factory | Exhibition hall |
| 3 | Royal Tapestry Factory | Factory | Museum |
| 4 | Isabel II Exhibition Hall | Water deposit | Exhibition center |
| 5 | Railway Museum of Madrid | Railway station | Museum |
| 6 | Atocha Station Tropical Garden | Railway station | Public garden |
| 7 | San Miguel Market | Market | Center for contemporary culture |
| 8 | Joaquín Leguina Regional Library | Brewery | Library and exhibition center |
| 9 | Motor Warehouse/Andén 0 | Power station | Exhibition center |
| 10 | Caixa Forum | Power station | Exhibition center |
| 11 | Medialab Prado | Sawmill | Center for contemporary culture |
| 12 | ABC Museum | Brewery | Exhibition center |
| 13 | Canal Foundation | Water deposit | Exhibition center |
| 14 | Circo Price Theater | Food factory | Center for contemporary culture |
| 15 | Matadero Madrid | Slaughterhouse | Center for contemporary culture |
| 16 | La Neomudéjar | Workshops | Center for contemporary culture |
| 17 | Chamberí Station/Andén 0 | Metro station | Center for contemporary culture |

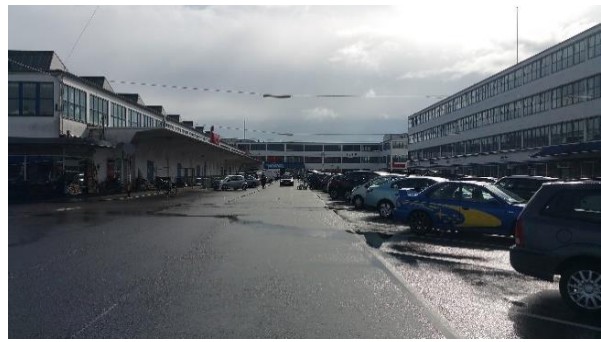

<div align="center">Meatpacking District</div>

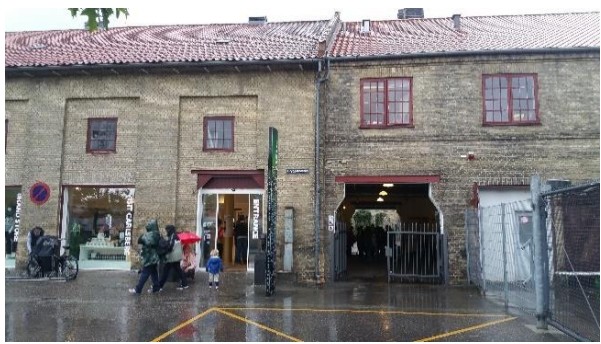

<div align="center">Carlsberg Brewery</div>

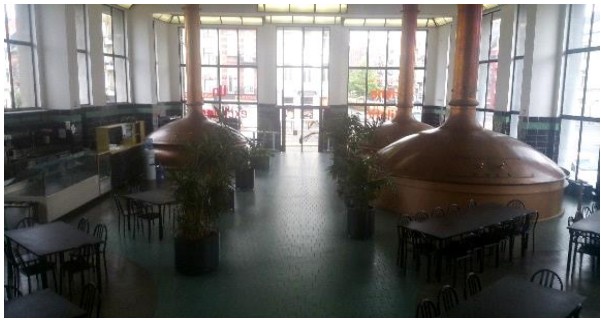

<div align="center">Wiels Art Center</div>

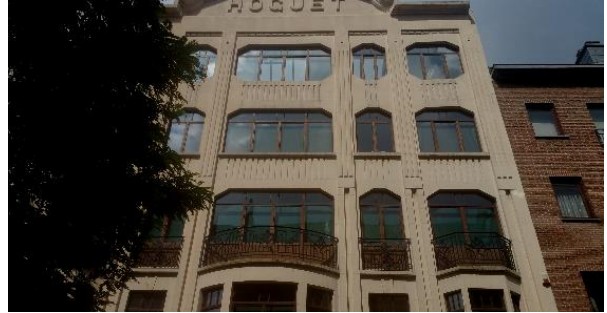

<div align="center">Book House</div>

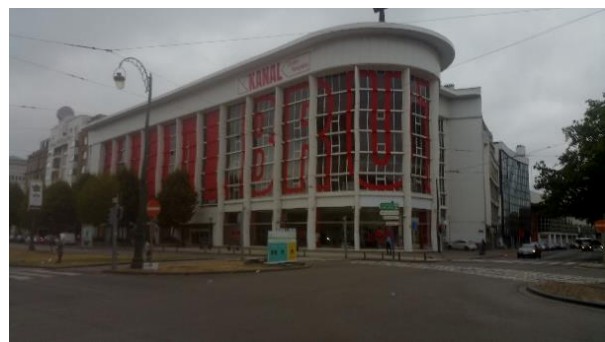

<div align="center">Kanal-Centre Pompidou</div>

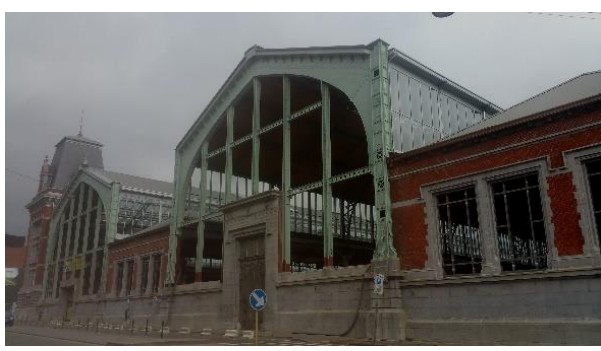

<div align="center">Tour et Taxis</div>

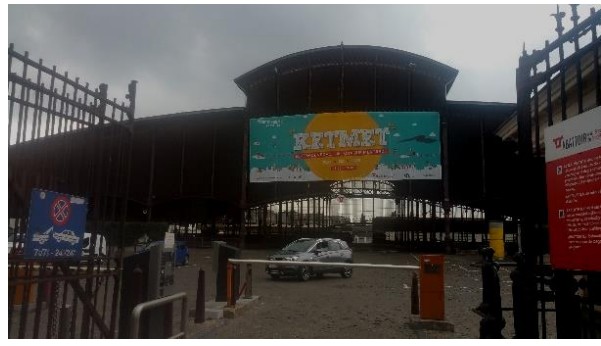

<div align="center">Abbatoir</div>

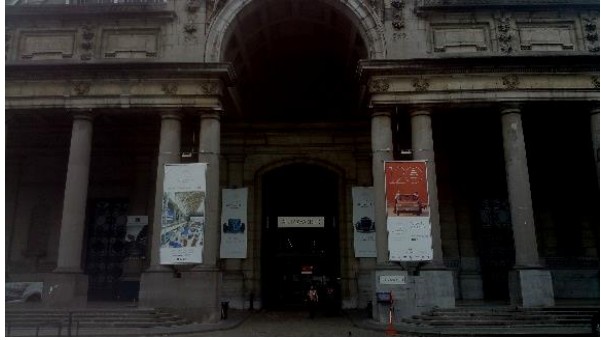

<div align="center">Autoworld</div>

**Figure A1.** Examples of cultural spaces of industrial origin in Copenhagen and Brussels. Source: the authors.

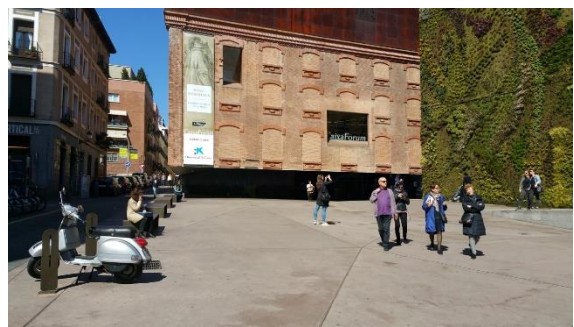

Caixa Forum

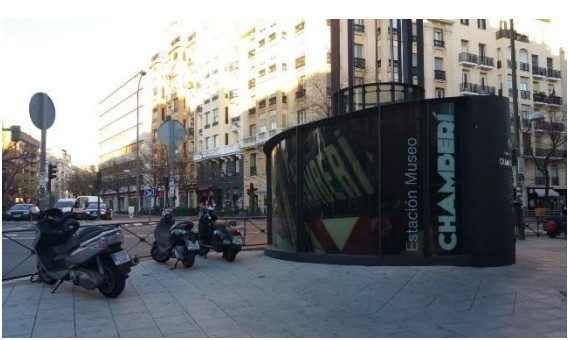

Chamberí Station/Andén 0

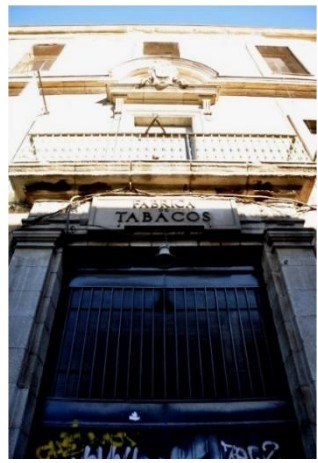

CSA La Tabacalera de Lavapiés

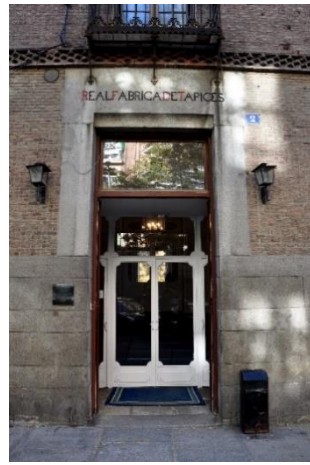

Royal Tapestry Factory

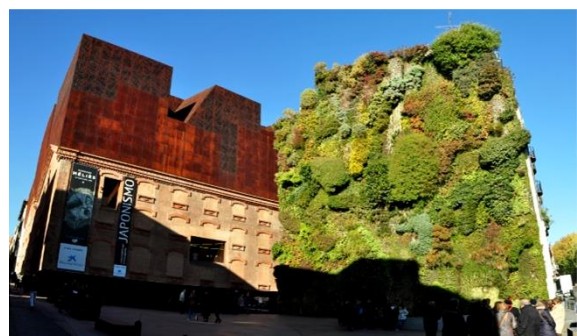

Caixa Forum

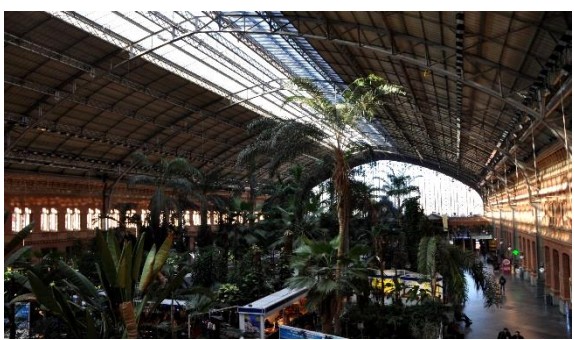

Tropical Garden Atocha Station

**Figure A2.** Examples of the cultural spaces of industrial origin in Madrid. Source: the authors.

**Table A4.** Coefficient of determination (Pearson).

| Variables | Cultural Representativeness | Tourist Use |
|---|---|---|
| Cultural representativeness | 1 | 0.083 |
| Tourist use | 0.083 | 1 |

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
