# Peer review of "Urban Industrial Tourism: Cultural Sustainability as a Tool for Confronting Overtourism—Cases of Madrid, Brussels, and Copenhagen"

_sustainability, doi:10.3390/su13094694_

Round 1

Reviewer 1 Report

Dear Authors,

Thank you for reading this interesting research. In my opinion, significant changes should be made before publication.

  1. The abstract needs to be improved. Currently, it does not reflect the full problem of work, research results and methodology.
  2. The introduction is very short, there is no indication of the research gap and the purpose of the research.
  3. The methods section in its current form is not very clear. It is not clear where the data came from, on what basis the weights were given in tables 5, etc.
  4. Section 4 is way too long. Some information from 4.1. can be included in the introduction and in the justification for the selection of objects
  5. Part of the description in sections 4.2, 4.3 and 4.4 is a methodology - a selection of cultural spaces of industrial origin.
  6. The results section provides a simple analysis. I propose to extend this section to use more advanced statistical methods.

Author Response

Esteemed referee,

Thank you very much for your recommendations and suggestions, which have contributed to the scientific improvement of the article. Our responses to and clarifications based on your observations are as follow:

1.    The abstract needs to be improved. Currently, it does not reflect the full problem of work, research results and methodology.

•    The Abstract has been modified to address the suggestions made. See attached file. 

2.    The introduction is very short, there is no indication of the research gap and the purpose of the research.

•    The Introduction has been expanded to clearly indicate the research gap as well as the purpose of our research. See attached file. 

3.    The methods section in its current form is not very clear. It is not clear where the data came from, on what basis the weights were given in tables 5, etc.

•    To give the Methods section a clearer structure, the text has been simplified and a justification for our selection of the case studies has been included. 
•    We have added the sentence “The evaluation criteria of this scale have been debated, agreed upon, and rated by the researchers and are listed in Table 2” to explain how the data was obtained. 

4.    Section 4 is way too long. Some information from 4.1. can be included in the introduction and in the justification for the selection of objects.

•    Section 4 has been shortened. The information from point 4.1 has been moved to the Methods section to justify the selection of the case studies.
•    Section 4.1 (‘A brief characterization of the industrial heritage in the selected cases’) now contains information previously found in sections 4.2 to 4.4, with the purpose of clarifying the matrices. 

5.    Part of the description in sections 4.2, 4.3 and 4.4 is a methodology - a selection of cultural spaces of industrial origin.

•    Section 4 has been shortened. The information from point 4.1 has been moved to the Methods section to justify the selection of the case studies.

6.    The results section provides a simple analysis. I propose to extend this section to use more advanced statistical methods.

•    Although we value this recommendation and understand that it would improve the article, it is at present beyond our possibilities. We will try to apply advanced statistical methods in future research. To improve upon the Results section, we have included a complementary Conclusions section that in our estimation closes the paper in a more satisfactory way. 

Reviewer 2 Report

This is a good article which needs to be considered for publication in Sustainability. A minor revision is needed following the next four issues:

1) The introduction needs to better highlight the aims of the study and to what current theoretical literature it brings value or additions; 

2) The literature review on urban industrial tourism and cultural sustainability for confronting overtourism could be a little bit updated. For instance, authors can mention that cultural cities could bring a lot of money from tourism activities, see Light Duncan et al.'s work on changing tourism in post-communist European cities (doi: 10.1080/19448953.2020.1775405) or see Vesalon et al's work in Journal of Urban and Regional Analysis, 2019 on how Little Vienna strives hard to become an urban tourism city or an avant-garde city. Then Alberto Vanolo's study on Turin should be mentioned for creative and branding/tourism cities, see 10.1016/j.cities.2008.08.001, while another important study on urban tourism or creative cities is K. Bodirsky's study on the city of Berlin published in International Journal of Cultural Policy, 2012.  Authors can also mention that some local city tensions regarding heritage buildings and potential corruption of the local elites (see doi: 10.1111/1468-2427.12775, doi: 10.1080/02723638.2019.1664252 and doi: 10.1080/02723638.2018.1472444) could create problems for the authorities' desire for certain cities to become more touristically reknown, mainly when certain cities fight for European capital of culture competitions.

3) The paper should say several lines on the limitations of this study. What are the limits of the data presented in this study?

4) There are no conclusions in this paper, authors preferring to include one concluding paragraph in the end of the discussion. I would suggest the authors to make two separate sections: discussions which should relate the findings of the paper with the literature review and conclusions which will show the novelty of the findings and how they advance current theories on urban industrial heritage and tourism.

Finally, in the conclusions authors can remind the readers that COVID19 impacted on mass tourism and can mention some highly viewed and cited articles dealing with COVID19 and urban development/transnational labours/tourism aspects (see doi: 10.1080/15387216.2020.1780929 and see Qiu et al's article in Annals of Tourism Research - doi: 10.1016/j.annals.2020.102994).

The excellent merit of this paper is its very good method and data interpretation for which I would like to congratulate the authors.

Author Response

Esteemed referee,

Thank you very much for your recommendations and suggestions, which have contributed to the scientific improvement of the article. Our responses to and clarifications based on your observations are as follow:

1) The introduction needs to better highlight the aims of the study and to what current theoretical literature it brings value or additions.

•    The Introduction has been expanded to indicate the objectives of the study and the paper’s theoretical frame of reference. See attached file. 

2) The literature review on urban industrial tourism and cultural sustainability for confronting overtourism could be a little bit updated. For instance, authors can mention that cultural cities could bring a lot of money from tourism activities, see Light Duncan et al.'s work on changing tourism in post-communist European cities (doi: 10.1080/19448953.2020.1775405) or see Vesalon et al's work in Journal of Urban and Regional Analysis, 2019 on how Little Vienna strives hard to become an urban tourism city or an avant-garde city. Then Alberto Vanolo's study on Turin should be mentioned for creative and branding/tourism cities, see 10.1016/j.cities.2008.08.001, while another important study on urban tourism or creative cities is K. Bodirsky's study on the city of Berlin published in International Journal of Cultural Policy, 2012.  Authors can also mention that some local city tensions regarding heritage buildings and potential corruption of the local elites (see doi: 10.1111/1468-2427.12775, doi: 10.1080/02723638.2019.1664252 and doi: 10.1080/02723638.2018.1472444) could create problems for the authorities' desire for certain cities to become more touristically reknown, mainly when certain cities fight for European capital of culture competitions.

•    Thank you very much for the bibliographic recommendations, which we have found very useful. We have incorporated some, including:

1.    Qui, R.; Park.; Li, S.; Song.; H. Social costs of tourism during the COVID-19 pandemic. Annals of Tourism Research, 2020, 84.
2.    Vanolo, A. The image of the creative city: some reflections on urban branding in Turin. Cities. 2008, 25(6), 370-382.
3.    Novy, J.; Colomb, C. Struggling for the Right to the (Creative) City in Berlin and Hamburg: New Urban Social Movements, New ‘Spaces of Hope’? International Journal of Urban and Regional Research. 2013, 37(5), 1816-1838.  
4.    Light, D.; CreÅ£an, R.; Voiculescu, S.; Sebastina Jucu, I. Introduction: Changing Tourism in the Cities of Post-communist Central and Eastern Europe. Journal of Balkan and Near Eastern Studies. 2020, 22(4), 465-477.

•    We have also incorporated references based on the topics suggested, especially ‘COVID-19 and tourism’: 

1.    Cañada, E.; Murray, I. Turistificación confinada. Alba Sud Editorial: Barcelona, Spain, 2021.
2.    Simancas, M.; Hernández, R.; Padrón, N. Turismo pos-COVID-19. Reflexiones, retos y oportunidades. Cátedra de Turismo Caja Canarias-Ashotel. Universidad de la Laguna: La Laguna, Spain, 2020.

3) The paper should say several lines on the limitations of this study. What are the limits of the data presented in this study?

•    Limitations of the study have been incorporated into the Introduction:

Industrial heritage is a fairly recent discipline, scarcely 70 years old; and while the ability of industrial heritage to attract tourist flows has prompted more interested over the past decade, studies remain few. It is therefore difficult to compare our results with those of similar investigations. Although this research certainly contributes to covering a gap in this area of knowledge, the number of cities analyzed should be increased to obtain more viable patterns of behavior related to urban industrial tourism from a technical/scientific perspective. In addition to this limitation, it must be noted that the COVID-19 pandemic has altered the orientation of this study. This unusual but persistent circumstance has necessarily become a factor in our analysis, given that the scenario created by the pandemic has introduced an element of uncertainty that could conceivably compromise the validity of the data over the long term. 

4) There are no conclusions in this paper, authors preferring to include one concluding paragraph in the end of the discussion. I would suggest the authors to make two separate sections: discussions which should relate the findings of the paper with the literature review and conclusions which will show the novelty of the findings and how they advance current theories on urban industrial heritage and tourism.

•    To address this suggestion, a Conclusions section has been introduced, separate from the Results and Discussion. See attached file. 

Reviewer 3 Report

Please find below comments and suggestions on paper substance and structure.

Several weaknesses exist: The paper addresses an interesting topic, the one of impact of over tourist on selected European cities and tourism transformations based on the cultural resources. Authors prove to be experts in sustainable cultural tourism and tourism development; and their English language level is also very good.

Connection of individual parts into a coherent research (title-aim-objectives-methods-results)

Abstract

 it is not clear, need to be rewritten in accordance with conventions (informative abstract about aims and research results). It is recommended to better focus „key words“ on the words that are connected to the research structure rather that those in the title (sustainability, tourism sustainability, cultural sustainability).

Introduction

 is mixed with literature review. Conceptual framework is missing. Author(s) need to separate the introduction (research problem background, research problem announcement and justification, study objectives/ hypothesis, a short paragraph on methods and procedures, paper structure; from literature review. This section may end with a (optionally) conceptual framework. Indicate the actuality of the research problem (eg. according to changes in global tourism trends, specifics of tourism demand, mass vs alternative tourism, cultural tourism).

Theoretical part

This section may end with a theoretical relationship model (as a part of authors' conclucion according to literature review). It is also suggestion to shorten the literature review. It is too long.

Discussion/Conclusion

Discussion is based on findings. Please develop conclusions. The paper should be completed with conclusions and comments/suggestions about future researches, separately from discussion. Please develop conclusions based on Your findings (without citations) and clearly point study limitations.

Author Response

Esteemed referee,

Thank you very much for your recommendations and suggestions, which have contributed to the scientific improvement of the article. Our responses to and clarifications based on your observations are as follow:

Abstract. it is not clear, need to be rewritten in accordance with conventions (informative abstract about aims and research results). It is recommended to better focus „key words“ on the words that are connected to the research structure rather that those in the title (sustainability, tourism sustainability, cultural sustainability).

•    The Abstract has been modified to address the suggestions made. See attached file. 
•    The keywords have been reoriented based on the suggestions: overtourism; sustainability; industrial heritage; industrial tourism; new urban spaces, cultural resources.

Introduction is mixed with literature review. Conceptual framework is missing. Author(s) need to separate the introduction (research problem background, research problem announcement and justification, study objectives/ hypothesis, a short paragraph on methods and procedures, paper structure; from literature review. This section may end with a (optionally) conceptual framework. Indicate the actuality of the research problem (eg. according to changes in global tourism trends, specifics of tourism demand, mass vs alternative tourism, cultural tourism).

•    The Introduction has been expanded to indicate the research problem background, the research problem announcement and justification, the study objectives / hypothesis, and the methods and procedures.

Theoretical part. This section may end with a theoretical relationship model (as a part of authors' conclusion according to literature review). It is also suggestion to shorten the literature review. It is too long.

•    The literature review has been shortened. 

Discussion/Conclusion. Discussion is based on findings. Please develop conclusions. The paper should be completed with conclusions and comments/suggestions about future researches, separately from discussion. Please develop conclusions based on Your findings (without citations) and clearly point study limitations.

•    To address this suggestion, a Conclusions section has been introduced separate from the Results and Discussion. Future lines of research have been indicated, both in the Introduction (related to the limitations) and in the new Conclusions section.  

Reviewer 4 Report

The papper is innovative, interesting and worht reading. Would like to recommend the authors to insert in the text the reason they choose the Tripadvisor platform as a source of information to qualify the industrials sites. 

Author Response

Esteemed referee,

Thank you very much for your very positive comments and evaluation. We are pleased by your interest in our research and we hope you like the revised version equally well.

Reviewer 5 Report

I enjoyed reading this, though as a qualitative researcher, a lot of the quantitative components lost me at points. The conversion of industrial sites to tourist spots is clearly an important and highly relevant subject, and you all seem very qualified and well-sourced in approaching it. However, I think it would help to get to your thesis before 3.1, and define 'Industrial tourism' quicker. Also, I may have possibly missed it, but how did you quantify cultural representativeness? And what did you mean by "Own elaboration" on the charts? 

Personally, I have the most familiarity with Madrid, and I have been to Brussels once (so I wouldn't say I remember much about it). It is clear that your base in Madrid takes precedence here, making the research a bit lopsided, though I think Brussels and Copenhagen do add something. There are a few typos ("Copenhage" on p9 map, p13 chart "meatpaacking"). Also, did you address that dip in that chart on 7 for Brussels? Was it because of the terror attack?

Also, I found the heavy use of the Passive Voice distracting - perhaps shifting into the active voice may help provide a background to how you arrived at these measures and observations on tourist traffic and cultural representativeness, etc. 

Author Response

Esteemed referee,

Thank you very much for your recommendations and suggestions, which have contributed to the scientific improvement of the article. Our responses to and clarifications based on your observations are as follow:

I think it would help to get to your thesis before 3.1, and define 'Industrial tourism' quicker. 
•    We have expanded the introduction by indicating our working hypothesis and making reference to the concept of industrial tourism. 
,

Also, I may have possibly missed it, but how did you quantify cultural representativeness?
•    The research team, with expertise in both cultural heritage and industrial heritage, designed and agreed upon a scoring scale from 1 to 3 (Likert scale). The team also reached consensus on the scores given to each cultural asset. The Methodology section specifies our process. 

And what did you mean by "Own elaboration" on the charts? 
•    We use the term "Own elaboration" as the standard (translated from Spanish) where the authors have composed their own tables, cartography, photographs, etc. If you consider different wording to be more suitable, please do not hesitate to tell us. 

Personally, I have the most familiarity with Madrid, and I have been to Brussels once (so I wouldn't say I remember much about it). It is clear that your base in Madrid takes precedence here, making the research a bit lopsided, though I think Brussels and Copenhagen do add something. There are a few typos ("Copenhage" on p9 map, p13 chart "meatpaacking"). 
•    The typos indicated have been corrected, thank you for catching them. 

Also, did you address that dip in that chart on 7 for Brussels? Was it because of the terror attack?
•    It’s really not within our area of expertise to respond to that, but it would be an interesting topic for an article. 

Also, I found the heavy use of the Passive Voice distracting - perhaps shifting into the active voice may help provide a background to how you arrived at these measures and observations on tourist traffic and cultural representativeness, etc. 
•    We have tried to make the text more dynamic. 

Round 2

Reviewer 1 Report

Dear Authors,

The manuscript has been significantly improved. The revised version is satisfactory.